# #EXPLORATION:
# A STUDY OF COUNT-BASED EXPLORATION
# FOR DEEP REINFORCEMENT LEARNING

**Haoran Tang**[1*]**, Rein Houthooft**[3,4*]**, Davis Foote**[2]**, Adam Stooke**[2]**, Xi Chen**[2,4]**,**
**Yan Duan**[2,4]**, John Schulman**[4]**, Filip De Turck**[3]**, Pieter Abbeel** [2,4]
[1] UC Berkeley, Department of Mathematics
[2] UC Berkeley, Department of Electrical Engineering and Computer Sciences
[3] Ghent University – imec, Department of Information Technology
[4] OpenAI

## ABSTRACT

Count-based exploration algorithms are known to perform near-optimally when used in conjunction with tabular reinforcement learning (RL) methods for solving small discrete Markov decision processes (MDPs). It is generally thought that count-based methods cannot be applied in high-dimensional state spaces, since most states will only occur once. Recent deep RL exploration strategies are able to deal with high-dimensional continuous state spaces through complex heuristics, often relying on *optimism in the face of uncertainty* or *intrinsic motivation*. In this work, we describe a surprising finding: a simple generalization of the classic count-based approach can reach near state-of-the-art performance on various high-dimensional and/or continuous deep RL benchmarks. States are mapped to hash codes, which allows to count their occurrences with a hash table. These counts are then used to compute a reward bonus according to the classic count-based exploration theory. We find that simple hash functions can achieve surprisingly good results on many challenging tasks. Furthermore, we show that a domain-dependent learned hash code may further improve these results. Detailed analysis reveals important aspects of a good hash function: 1) having appropriate granularity and 2) encoding information relevant to solving the MDP. This exploration strategy achieves near state-of-the-art performance on both continuous control tasks and Atari 2600 games, hence providing a simple yet powerful baseline for solving MDPs that require considerable exploration.

## 1 INTRODUCTION

Reinforcement learning (RL) studies an agent acting in an initially unknown environment, learning through trial and error to maximize rewards. It is impossible for the agent to act near-optimally until it has sufficiently explored the environment and identified all of the opportunities for high reward, in all scenarios. A core challenge in RL is how to balance exploration—actively seeking out novel states and actions that might yield high rewards and lead to long-term gains; and exploitation—maximizing short-term rewards using the agent's current knowledge. While there are exploration techniques for finite MDPs that enjoy theoretical guarantees, there are no fully satisfying techniques for high-dimensional state spaces; therefore, developing more general and robust exploration techniques is an active area of research.

Most of the recent state-of-the-art RL results have been obtained using simple exploration strategies such as uniform sampling (Mnih et al., 2015) and i.i.d./correlated Gaussian noise (Schulman et al., 2015; Lillicrap et al., 2015). Although these heuristics are sufficient in tasks with well-shaped rewards, the sample complexity can grow exponentially (with state space size) in tasks with sparse rewards (Osband et al., 2016b). Recently developed exploration strategies for deep RL have led to significantly improved performance on environments with sparse rewards. Bootstrapped DQN

---

*These authors contributed equally.

(Osband et al., 2016a) led to faster learning in a range of Atari 2600 games by training an ensemble of Q-functions. Intrinsic motivation methods using pseudo-counts achieve state-of-the-art performance on Montezuma's Revenge, an extremely challenging Atari 2600 game (Bellemare et al., 2016). Variational Information Maximizing Exploration (VIME, Houthooft et al. (2016)) encourages the agent to explore by acquiring information about environment dynamics, and performs well on various robotic locomotion problems with sparse rewards. However, we have not seen a very simple and fast method that can work across different domains.

Some of the classic, theoretically-justified exploration methods are based on counting state-action visitations, and turning this count into a bonus reward. In the bandit setting, the well-known UCB algorithm of Lai & Robbins (1985) chooses the action $a_t$ at time $t$ that maximizes $\hat{r}(a_t) + \sqrt{\frac{2 \log t}{n(a_t)}}$ where $\hat{r}(a_t)$ is the estimated reward, and $n(a_t)$ is the number of times action $a_t$ was previously chosen. In the MDP setting, some of the algorithms have similar structure, for example, Model Based Interval Estimation–Exploration Bonus (MBIE-EB) of Strehl & Littman (2008) counts state-action pairs with a table $n(s, a)$ and adding a bonus reward of the form $\frac{\beta}{\sqrt{n(s,a)}}$ to encourage exploring less visited pairs. Kolter & Ng (2009) show that the inverse-square-root dependence is optimal. MBIE and related algorithms assume that the augmented MDP is solved analytically at each timestep, which is only practical for small finite state spaces.

This paper presents a simple approach for exploration, which extends classic counting-based methods to high-dimensional, continuous state spaces. We discretize the state space with a hash function and apply a bonus based on the state-visitation count. The hash function can be chosen to appropriately balance generalization across states, and distinguishing between states. We select problems from rllab (Duan et al., 2016) and Atari 2600 (Bellemare et al., 2012) featuring sparse rewards, and demonstrate near state-of-the-art performance on several games known to be hard for naïve exploration strategies. The main strength of the presented approach is that it is fast, flexible and complementary to most existing RL algorithms.

In summary, this paper proposes a generalization of classic count-based exploration to high-dimensional spaces through hashing (Section 2); demonstrates its effectiveness on challenging deep RL benchmark problems and analyzes key components of well-designed hash functions (Section 3).

## 2 METHODOLOGY

### 2.1 NOTATION

This paper assumes a finite-horizon discounted Markov decision process (MDP), defined by $(\mathcal{S}, \mathcal{A}, \mathcal{P}, r, \rho_0, \gamma, T)$, in which $\mathcal{S}$ is the state space, $\mathcal{A}$ the action space, $\mathcal{P}$ a transition probability distribution, $r : \mathcal{S} \times \mathcal{A} \rightarrow \mathbb{R}_{\geq 0}$ a reward function, $\rho_0$ an initial state distribution, $\gamma \in (0, 1]$ a discount factor, and $T$ the horizon. The goal of RL is to maximize the total expected discounted reward $\mathbb{E}_{\pi, \mathcal{P}} \left[ \sum_{t=0}^{T} \gamma^t r(s_t, a_t) \right]$ over a policy $\pi$, which outputs a distribution over actions given a state.

### 2.2 COUNT-BASED EXPLORATION VIA STATIC HASHING

Our approach discretizes the state space with a hash function $\phi : \mathcal{S} \rightarrow \mathbb{Z}$. An exploration bonus is added to the reward function, defined as

$$r^+(s, a) = \frac{\beta}{\sqrt{n(\phi(s))}}, \tag{1}$$

where $\beta \in \mathbb{R}_{\geq 0}$ is the bonus coefficient. Initially the counts $n(\cdot)$ are set to zero for the whole range of $\phi$. For every state $s_t$ encountered at time step $t$, $n(\phi(s_t))$ is increased by one. The agent is trained with rewards $(r + r^+)$, while performance is evaluated as the sum of rewards without bonuses.

Note that our approach is a departure from count-based exploration methods such as MBIE-EB since we use a state-space count $n(s)$ rather than a state-action count $n(s, a)$. State-action counts $n(s, a)$ are investigated in Appendix A.6, but no significant performance gains over state counting could be witnessed.

---

**Algorithm 1:** Count-based exploration through static hashing

---

1 Define state preprocessor $g : \mathcal{S} \rightarrow \mathbb{R}^K$

2 (In case of SimHash) Initialize $A \in \mathbb{R}^{k \times K}$ with entries drawn i.i.d. from the standard Gaussian distribution $\mathcal{N}(0, 1)$

3 Initialize a hash table with values $n(\cdot) \equiv 0$

4 **for** each iteration $j$ **do**

5 Collect a set of state-action samples $\{(s_m, a_m)\}_{m=0}^{M}$ with policy $\pi$

6 Compute hash codes through any LSH method, e.g., for SimHash, $\phi(s_m) = \operatorname{sgn}(Ag(s_m))$

7 Update the hash table counts $\forall m : 0 \leq m \leq M$ as $n(\phi(s_m)) \leftarrow n(\phi(s_m)) + 1$

8 Update the policy $\pi$ using rewards $\left\{ r(s_m, a_m) + \dfrac{\beta}{\sqrt{n(\phi(s_m))}} \right\}_{m=0}^{M}$ with any RL algorithm

---

Clearly the performance of this method will strongly depend on the choice of hash function $\phi$. One important choice we can make regards the *granularity* of the discretization: we would like for "distant" states to be be counted separately while "similar" states are merged. If desired, we can incorporate prior knowledge into the choice of $\phi$, if there would be a set of salient state features which are known to be relevant.

Algorithm 1 summarizes our method. The main idea is to use locality-sensitive hashing (LSH) to convert continuous, high-dimensional data to discrete hash codes. LSH is a popular class of hash functions for querying nearest neighbors based on certain similarity metrics (Andoni & Indyk, 2006). A computationally efficient type of LSH is SimHash (Charikar, 2002), which measures similarity by angular distance. SimHash retrieves a binary code of state $s \in \mathcal{S}$ as

$$\phi(s) = \operatorname{sgn}(Ag(s)) \in \{-1, 1\}^k, \tag{2}$$

where $g : \mathcal{S} \rightarrow \mathbb{R}^d$ is an optional preprocessing function and $A$ is a $k \times d$ matrix with i.i.d. entries drawn from a standard Gaussian distribution $\mathcal{N}(0, 1)$. The value for $k$ controls the granularity: higher values lead to fewer collisions and are thus more likely to distinguish states.

## 2.3 Count-Based Exploration via Learned Hashing

When the MDP states have a complex structure, as is the case with image observations, measuring their similarity directly in pixel space fails to provide the semantic similarity measure one would desire. Previous work in computer vision (Lowe, 1999; Dalal & Triggs, 2005; Tola et al., 2010) introduce manually designed feature representations of images that are suitable for semantic tasks including detection and classification. More recent methods learn complex features directly from data by training convolutional neural networks (Krizhevsky et al., 2012; Simonyan & Zisserman, 2014; He et al., 2015). Considering these results, it may be difficult for SimHash to cluster states appropriately using only raw pixels.

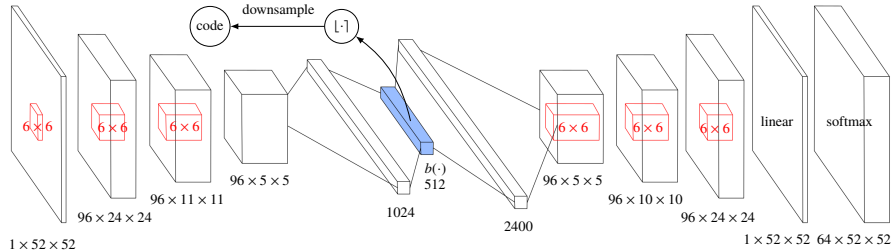

Figure 1: The autoencoder (AE) architecture; the solid block represents the dense sigmoidal binary code layer, after which noise $U(-a, a)$ is injected.

Therefore, we propose to use an autoencoder (AE) consisting of convolutional, dense, and transposed convolutional layers to learn meaningful hash codes in one of its hidden layers. This AE takes as input states $s$ and contains one special dense layer comprised of $K$ saturating activation functions,

---

**Algorithm 2:** Count-based exploration using learned hash codes

1  Define state preprocessor $g : \mathcal{S} \to \mathbb{B}^K$ as the binary code resulting from the autoencoder (AE)
2  Initialize $A \in \mathbb{R}^{k \times K}$ with entries drawn i.i.d. from the standard Gaussian distribution $\mathcal{N}(0, 1)$
3  Initialize a hash table with values $n(\cdot) \equiv 0$
4  **for** each iteration $j$ **do**
5 Collect a set of state-action samples $\{(s_m, a_m)\}_{m=0}^{M}$ with policy $\pi$
6 Add the state samples $\{s_m\}_{m=0}^{M}$ to a FIFO replay pool $\mathcal{R}$
7 **if** $j \bmod j_{\text{update}} = 0$ **then**
8 Update the AE loss function in Eq. (3) using samples drawn from the replay pool
 $\{s_n\}_{n=1}^{N} \sim \mathcal{R}$, for example using stochastic gradient descent
9 Compute $g(s_m) = \lfloor b(s_m) \rceil$, the $K$-dim rounded hash code for $s_m$ learned by the AE
10 Project $g(s_m)$ to a lower dimension $k$ via SimHash as $\phi(s_m) = \operatorname{sgn}(Ag(s_m))$
11 Update the hash table counts $\forall m : 0 \leq m \leq M$ as $n(\phi(s_m)) \leftarrow n(\phi(s_m)) + 1$
12 Update the policy $\pi$ using rewards $\left\{ r(s_m, a_m) + \dfrac{\beta}{\sqrt{n(\phi(s_m))}} \right\}_{m=0}^{M}$ with any RL algorithm

---

more specifically sigmoid functions. By rounding the sigmoid output $b(s)$ of this layer to the closest binary number, any state $s$ can be binarized.

Since gradients cannot be back-propagated through a rounding function, an alternative method must be used to ensure that distinct states are mapped to distinct binary codes. Therefore, uniform noise $U(-a, a)$ is added to the sigmoid output. By choosing uniform noise with a sufficiently high variance, the AE is only capable of reconstructing distinct inputs $s$ if its hidden dense layer outputs values $b(s)$ that are sufficiently far apart from each other (Gregor et al., 2016). Feeding a state $s$ to the AE input, extracting $b(s)$ and rounding it to $\lfloor b(s) \rceil$ yields a learned binary code. As such, the loss function $L(\cdot)$ over a set of collected states $\{s_i\}_{i=1}^{N}$ is defined as

$$L\left(\{s_n\}_{n=1}^{N}\right) = -\frac{1}{N} \sum_{n=1}^{N} \left[ \log p(s_n) - \frac{\lambda}{K} \sum_{i=1}^{K} \min\left\{ (1 - b_i(s_n))^2, b_i(s_n)^2 \right\} \right]. \tag{3}$$

This objective function consists of a cross-entropy term and a term that pressures the binary code layer to take on binary values, scaled by $\lambda \in \mathbb{R}_{\geq 0}$. The reasoning behind this is that uniform noise $U(-a, a)$ alone is insufficient, in case the AE does not use a particular sigmoid unit. This term ensures that an unused binary code output is assigned an arbitrary binary value. When omitting this term, the code is more prone to oscillations, causing unwanted bit flips, and destabilizing the counting process.

In order to make the AE train sufficiently fast—which is required since it is updated during the agent's training—we make use of a pixel-wise softmax output layer (van den Oord et al., 2016) that shares weights between all pixels. The different softmax outputs merge together pixel intensities into discrete bins. The architectural details are described in Appendix A.1 and are depicted in Figure 1. Because the code dimension often needs to be large in order to correctly reconstruct the input, we apply a downsampling procedure to the resulting binary code $\lfloor b(s) \rceil$, which can be done through random projection to a lower-dimensional space via SimHash as in Eq. (2).

One the one hand, it is important that the mapping from state to code needs to remain relatively consistent over time, which is nontrivial as the AE is constantly updated according to the latest data (Algorithm 2 line 8). An obvious solution would be to significantly downsample the binary code to a very low dimension, or by slowing down the training process. But on the other hand, the code has to remain relatively unique for states that are both distinct and close together on the image manifold. This is tackled both by the second term in Eq. (3) and by the saturating behavior of the sigmoid units. As such, states that are already well represented in the AE hidden layers tend to saturate the sigmoid units, causing the resulting loss gradients to be close to zero and making the code less prone to change.

## 3 EXPERIMENTS

Experiments were designed to investigate and answer the following research questions:

1. Can count-based exploration through hashing improve performance significantly across different domains? How does the proposed method compare to the current state of the art in exploration for deep RL?

2. What is the impact of learned or static state preprocessing on the overall performance when image observations are used?

3. What factors contribute to good performance, e.g., what is the appropriate level of granularity of the hash function?

To answer question 1, we run the proposed method on deep RL benchmarks (rllab and ALE) that feature sparse rewards, and compare it to other state-of-the-art algorithms. Question 2 is answered by trying out different image preprocessors on Atari 2600 games. Finally, we investigate question 3 in Section 3.3 and 3.4. Trust Region Policy Optimization (TRPO, Schulman et al. (2015)) is chosen as the RL algorithm for all experiments, because it can handle both discrete and continuous action spaces, it can conveniently ensure stable improvement in the policy performance, and is relatively insensitive to hyperparameter changes. The hyperparameters settings are reported in Appendix A.1.

## 3.1 CONTINUOUS CONTROL

The rllab benchmark (Duan et al., 2016) consists of various control tasks to test deep RL algorithms. We selected several variants of the basic and locomotion tasks that use sparse rewards, as shown in Figure 2, and adopt the experimental setup as defined in (Houthooft et al., 2016)—a description can be found in Appendix A.2. These tasks are all highly difficult to solve with naïve exploration strategies, such as adding Gaussian noise to the actions.

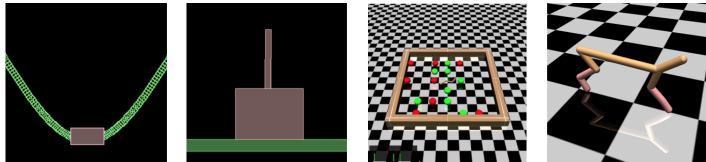

Figure 2: Illustrations of the rllab tasks used in the continuous control experiments, namely MountainCar, CartPoleSwingup, SimmerGather, and HalfCheetah; taken from (Duan et al., 2016).

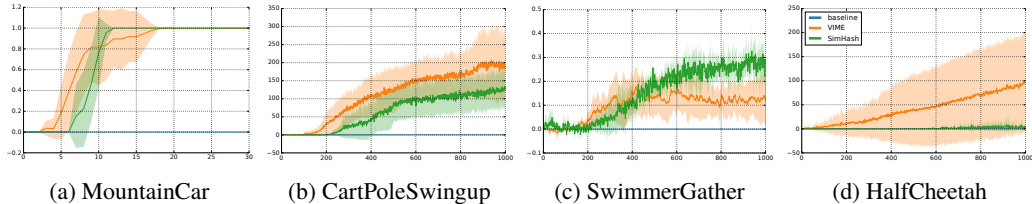

(a) MountainCar    (b) CartPoleSwingup    (c) SwimmerGather    (d) HalfCheetah

Figure 3: Mean average return of different algorithms on rllab tasks with sparse rewards; the solid line represents the mean average return, while the shaded area represents one standard deviation, over 5 seeds for the baseline and SimHash.

Figure 3 shows the results of TRPO (baseline), TRPO-SimHash, and VIME (Houthooft et al., 2016) on the classic tasks MountainCar and CartPoleSwingup, the locomotion task HalfCheetah, and the hierarchical task SwimmerGather. Using count-based exploration with hashing is capable of reaching the goal in all environments (which corresponds to a nonzero return), while baseline TRPO with Gaussian control noise fails completely. Although TRPO-SimHash picks up the sparse reward on HalfCheetah, it does not perform as well as VIME. In contrast, the performance of SimHash is comparable with VIME on MountainCar, while it outperforms VIME on SwimmerGather.

## 3.2 ARCADE LEARNING ENVIRONMENT

The Arcade Learning Environment (ALE, Bellemare et al. (2012)), which consists of Atari 2600 video games, is an important benchmark for deep RL due to its high-dimensional state space and wide

variety of games. In order to demonstrate the effectiveness of the proposed exploration strategy, six games are selected featuring long horizons while requiring significant exploration: Freeway, Frostbite, Gravitar, Montezuma's Revenge, Solaris, and Venture. The agent is trained for 500 iterations in all experiments, with each iteration consisting of 0.1 M steps (the TRPO batch size, corresponds to 0.4 M frames). Policies and value functions are neural networks with identical architectures to (Mnih et al., 2016). Although the policy and baseline take into account the previous four frames, the counting algorithm only looks at the latest frame.

**BASS** To compare with the autoencoder-based learned hash code, we propose using Basic Abstraction of the ScreenShots (BASS, also called Basic; see Bellemare et al. (2012)) as a static preprocessing function $g$. BASS is a hand-designed feature transformation for images in Atari 2600 games. BASS builds on the following observations specific to Atari: 1) the game screen has a low resolution, 2) most objects are large and monochrome, and 3) winning depends mostly on knowing object locations and motions. We designed an adapted version of BASS[1], that divides the RGB screen into square cells, computes the average intensity of each color channel inside a cell, and assigns the resulting values to bins that uniformly partition the intensity range $[0, 255]$. Mathematically, let $C$ be the cell size (width and height), $B$ the number of bins, $(i, j)$ cell location, $(x, y)$ pixel location, and $z$ the channel.

$$\text{feature}(i, j, z) = \left\lfloor \frac{B}{255C^2} \sum_{(x,y) \in \text{cell}(i,j)} I(x, y, z) \right\rfloor . \tag{4}$$

Afterwards, the resulting integer-valued feature tensor is converted to an integer hash code ($\phi(s_t)$ in Line 6 of Algorithm 1). A BASS feature can be regarded as a miniature that efficiently encodes object locations, but remains invariant to negligible object motions. It is easy to implement and introduces little computation overhead. However, it is designed for generic Atari game images and may not capture the structure of each specific game very well.

Table 1: Atari 2600: average total reward after training for 50 M time steps. Boldface numbers indicate best results. Italic numbers are the best among our methods.

|  | Freeway | Frostbite[1] | Gravitar | Montezuma | Solaris | Venture |
|---|---|---|---|---|---|---|
| TRPO (baseline) | 16.5 | 2869 | 486 | 0 | 2758 | 121 |
| TRPO-pixel-SimHash | 31.6 | 4683 | 468 | 0 | 2897 | 263 |
| TRPO-BASS-SimHash | 28.4 | 3150 | *604* | *238* | 1201 | *616* |
| TRPO-AE-SimHash | ***33.5*** | ***5214*** | 482 | 75 | *4467* | 445 |
| Double-DQN | 33.3 | 1683 | 412 | 0 | 3068 | 98.0 |
| Dueling network | 0.0 | 4672 | 588 | 0 | 2251 | 497 |
| Gorila | 11.7 | 605 | **1054** | 4 | N/A | **1245** |
| DQN Pop-Art | 33.4 | 3469 | 483 | 0 | **4544** | 1172 |
| A3C+ | 27.3 | 507 | 246 | 142 | 2175 | 0 |
| pseudo-count[2] | 29.2 | 1450 | – | **3439** | – | 369 |

[1] While Vezhnevets et al. (2016) reported best score 8108, their evaluation was based on top 5 agents trained with 500M time steps, hence not comparable.
[2] Results reported only for 25 M time steps (100 M frames).

We compare our results to double DQN (van Hasselt et al., 2016b), dueling network (Wang et al., 2016), A3C+ (Bellemare et al., 2016), double DQN with pseudo-counts (Bellemare et al., 2016), Gorila (Nair et al., 2015), and DQN Pop-Art (van Hasselt et al., 2016a) on the "null op" metric[2]. We show training curves in Figure 4 and summarize all results in Table 1. Surprisingly, TRPO-pixel-SimHash already outperforms the baseline by a large margin and beats the previous best result on Frostbite. TRPO-BASS-SimHash achieves significant improvement over TRPO-pixel-SimHash on

---

[1]The original BASS exploits the fact that at most 128 colors can appear on the screen. Our adapted version does not make this assumption.

[2]The agent takes no action for a random number (within 30) of frames at the beginning of each episode.

Montezuma's Revenge and Venture, where it captures object locations better than other methods.[3] TRPO-AE-SimHash achieves near state-of-the-art performance on Freeway, Frostbite and Solaris.[4]

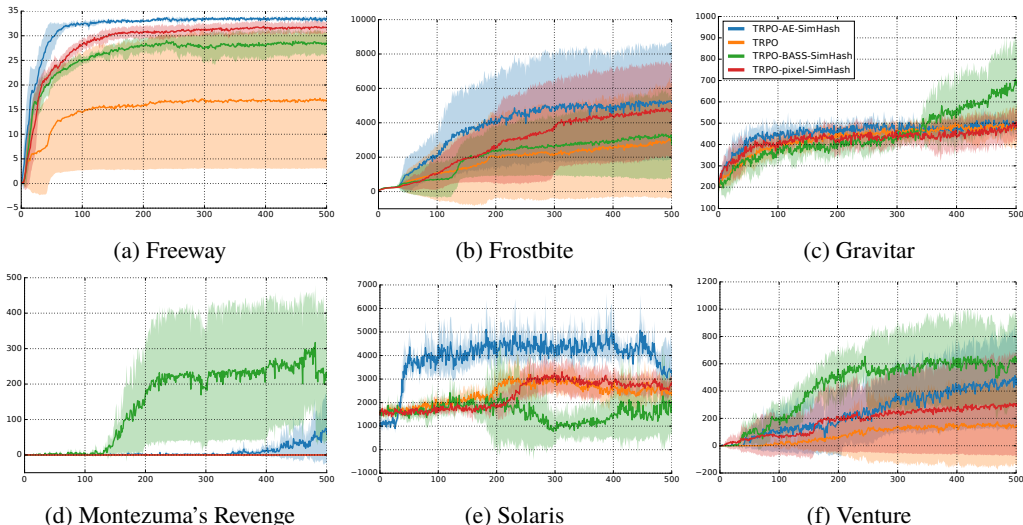

Figure 4: Atari 2600 games: the solid line is the mean average undiscounted return per iteration, while the shaded areas represent the one standard deviation, over 5 seeds for the baseline, TRPO-pixel-SimHash, and TRPO-BASS-SimHash, while over 3 seeds for TRPO-AE-SimHash.

As observed in Table 1, preprocessing images with BASS or using a learned hash code through the AE leads to much better performance on Gravitar, Montezuma's Revenge and Venture. Therefore, an static or adaptive preprocessing step can be important for a good hash function.

In conclusion, our count-based exploration method is able to achieve remarkable performance gains even with simple hash functions like SimHash on the raw pixel space. If coupled with domain-dependent state preprocessing techniques, it can sometimes achieve far better results.

### 3.3 GRANULARITY

While our proposed method is able to achieve remarkable results without requiring much tuning, the granularity of the hash function should be chosen wisely. Granularity plays a critical role in count-based exploration, where the hash function should cluster states without under-generalizing or over-generalizing. Table 2 summarizes granularity parameters for our hash functions. In Table 3 we summarize the performance of TRPO-pixel-SimHash under different granularities. We choose Frostbite and Venture on which TRPO-pixel-SimHash outperforms the baseline, and choose as reward bonus coefficient $\beta = 0.01 \times \frac{256}{k}$ to keep average bonus rewards at approximately the same scale. $k = 16$ only corresponds to 65536 distinct hash codes, which is insufficient to distinguish between semantically distinct states and hence leads to worse performance. We observed that $k = 512$ tends to capture trivial image details in Frostbite, leading the agent to believe that every state is new and equally worth exploring. Similar results are observed while tuning the granularity parameters for TRPO-BASS-SimHash and TRPO-AE-SimHash.

The best granularity depends on both the hash function and the MDP. While adjusting granularity parameter, we observed that it is important to lower the bonus coefficient as granularity is increased. This is because a higher granularity is likely to cause lower state counts, leading to higher bonus rewards that may overwhelm the true rewards.

---

[3] We provide videos of example game play and visualizations of the difference bewteen Pixel-SimHash and BASS-SimHash at https://www.youtube.com/playlist?list=PLAd-UMX6FkBQdLNWtY8nH1-pzYJA_1T55

[4] Note that some design choices in other algorithms also impact exploration, such as $\varepsilon$-greedy and entropy regularization. Nevertheless, it is still valuable to position our results within the current literature.

Table 2: Granularity parameters of various hash functions

| SimHash | $k$: size of the binary code |
|---|---|
| BASS | $C$: cell size; $B$: number of bins for each color channel |
| AE | $k$: down stream SimHash parameter; size of the binary code |
| | $\lambda$: binarization parameter |
| SmartHash | $s$: grid size for the agent's $(x, y)$ coordinates |

Table 3: Average score at 50 M time steps achieved by TRPO-pixel-SimHash

| $k$ | 16 | 64 | 128 | 256 | 512 |
|---|---|---|---|---|---|
| Frostbite | 3326 | 4029 | 3932 | **4683** | 1117 |
| Venture | 0 | 218 | 142 | 263 | **306** |

Table 4: Average score at 50 M time steps achieved by TRPO-SmartHash on Montezuma's Revenge (RAM observations)

| $s$ | 1 | 5 | 10 | 20 | 40 | 60 |
|---|---|---|---|---|---|---|
| score | 2598 | 2500 | **3533** | 3025 | 2500 | 1921 |

## 3.4 A Case Study of Montezuma's Revenge

Montezuma's Revenge is widely known for its extremely sparse rewards and difficult exploration (Bellemare et al., 2016). While our method does not outperform Bellemare et al. (2016) on this game, we investigate the reasons behind this through various experiments. The experiment process below again demonstrates the importance of a hash function having the correct granularity and encoding relevant information for solving the MDP.

Our first attempt is to use game RAM states instead of image observations as inputs to the policy (details in Appendix A.1), which leads to a game score of 2500 with TRPO-BASS-SimHash. Our second attempt is to manually design a hash function that incorporates domain knowledge, called *SmartHash*, which uses an integer-valued vector consisting of the agent's $(x, y)$ location, room number and other useful RAM information as the hash code (details in Appendix A.3). The best SmartHash agent is able to obtain a score of 3500. Still the performance is not optimal. We observe that a slight change in the agent's coordinates does not always result in a semantically distinct state, and thus the hash code may remain unchanged. Therefore we choose grid size $s$ and replace the $x$ coordinate by $\lfloor (x - x_{\min})/s \rfloor$ (similarly for $y$). The bonus coefficient is chosen as $\beta = 0.01\sqrt{s}$ to maintain the scale relative to the true reward[5] (see Table 4). Finally, the best agent is able to obtain 6600 total rewards after training for 1000 iterations (1000 M time steps), with a grid size $s = 10$.

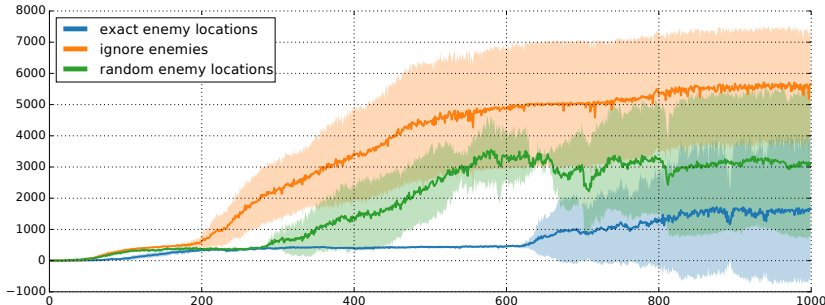

Figure 5: SmartHash results on Montezuma's Revenge (RAM observations): the solid line is the mean average undiscounted return per iteration, while the shaded areas represent the one standard deviation, over 5 seeds.

During our pursuit, we had another interesting discovery that the ideal hash function should not simply cluster states by their visual similarity, but instead by their relevance to solving the MDP. We

---

[5]The bonus scaling is chosen by assuming all states are visited uniformly and the average bonus reward should remain the same for any grid size.

experimented with including enemy locations in the first two rooms into SmartHash ($s = 10$), and observed that average score dropped to 1672 (at iteration 1000). Though it is important for the agent to dodge enemies, the agent also erroneously "enjoys" watching enemy motions at distance (since new states are constantly observed) and "forgets" that his main objective is to enter other rooms. An alternative hash function keeps the same entry "enemy locations", but instead only puts randomly sampled values in it, which surprisingly achieves better performance (3112). However, by ignoring enemy locations altogether, the agent achieves a much higher score (5661) (see Figure 5). In retrospect, we examine the hash codes generated by BASS-SimHash and find that codes clearly distinguish between visually different states (including various enemy locations), but fails to emphasize that the agent needs to explore different rooms. Again this example showcases the importance of encoding relevant information in designing hash functions.

## 4 RELATED WORK

Classic count-based methods such as MBIE (Strehl & Littman, 2005), MBIE-EB and (Kolter & Ng, 2009) solve an approximate Bellman equation as an inner loop before the agent takes an action (Strehl & Littman, 2008). As such, bonus rewards are propagated immediately throughout the state-action space. In contrast, contemporary deep RL algorithms propagate the bonus signal based on rollouts collected from interacting with environments, with value-based (Mnih et al., 2015) or policy gradient-based (Schulman et al., 2015; Mnih et al., 2016) methods, at limited speed. In addition, our proposed method is intended to work with contemporary deep RL algorithms, it differs from classical count-based method in that our method relies on visiting unseen states first, before the bonus reward can be assigned, making uninformed exploration strategies still a necessity at the beginning. Filling the gaps between our method and classic theories is an important direction of future research.

A related line of classical exploration methods is based on the idea of *optimism in the face of uncertainty* (Brafman & Tennenholtz, 2002) but not restricted to using counting to implement "optimism", e.g. R-Max (Brafman & Tennenholtz, 2002), UCRL (Jaksch et al., 2010), and $E^3$ (Kearns & Singh, 2002). These methods, similar to MBIE and MBIE-EB, have theoretical guarantees in tabular settings.

Bayesian RL methods (Kolter & Ng, 2009; Guez et al., 2014; Sun et al., 2011; Ghavamzadeh et al., 2015), which keep track of a distribution over MDPs, are an alternative to optimism-based methods. Extensions to continuous state space have been proposed by Pazis & Parr (2013) and Osband et al. (2016b).

Another type of exploration is curiosity-based exploration. These methods try to capture the agent's surprise about transition dynamics. As the agent tries to optimize for surprise, it naturally discovers novel states. We refer the reader to Schmidhuber (2010) and Oudeyer & Kaplan (2007) for an extensive review on curiosity and intrinsic rewards.

Several exploration strategies for deep RL have been proposed to handle high-dimensional state space recently. Houthooft et al. (2016) propose VIME, in which information gain is measured in Bayesian neural networks modeling the MDP dynamics, which is used an exploration bonus. Stadie et al. (2015) propose to use the prediction error of a learned dynamics model as an exploration bonus. Thompson sampling through bootstrapping is proposed by Osband et al. (2016a), using bootstrapped Q-functions.

The most related exploration strategy is proposed by Bellemare et al. (2016), in which an exploration bonus is added inversely proportional to the square root of a *pseudo-count* quantity. A state pseudo-count is derived from its log-probability improvement according to a density model over the state space, which in the limit converges to the empirical count. Our method is similar to pseudo-count approach in the sense that both methods are performing approximate counting to have the necessary generalization over unseen states. The difference is that a density model has to be designed and learned to achieve good generalization for pseudo-count whereas in our case generalization is obtained by a wide range of simple hash functions (not necessarily SimHash). Another interesting connection is that our method also implies a density model $\rho(s) = \frac{n(\phi(s))}{N}$ over all visited states, where $N$ is the total number of states visited. Another method similar to hashing is proposed by Abel et al. (2016), which clusters states and counts cluster centers instead of the true states, but this method has yet to be tested on standard exploration benchmark problems.

## 5 Conclusions

This paper demonstrates that a generalization of classical counting techniques through hashing is able to provide an appropriate signal for exploration, even in continuous and/or high-dimensional MDPs using function approximators, resulting in near state-of-the-art performance across benchmarks. It provides a simple yet powerful baseline for solving MDPs that require informed exploration.

## Acknowledgments

We would like to thank our colleagues at Berkeley and OpenAI for insightful discussions. This research was funded in part by ONR through a PECASE award. Yan Duan was also supported by a Berkeley AI Research lab Fellowship and a Huawei Fellowship. Xi Chen was also supported by a Berkeley AI Research lab Fellowship. We gratefully acknowledge the support of the NSF through grant IIS-1619362 and of the ARC through a Laureate Fellowship (FL110100281) and through the ARC Centre of Excellence for Mathematical and Statistical Frontiers. Adam Stooke gratefully acknowledges funding from a Fannie and John Hertz Foundation fellowship. Rein Houthooft is supported by a Ph.D. Fellowship of the Research Foundation - Flanders (FWO).

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

# A  APPENDICES

## A.1  HYPERPARAMETER SETTINGS

For the rllab experiments, we used batch size 5000 for all tasks except SwimmerGather, for which we used batch size 50000. CartpoleSwingup makes use of a neural network policy with one layer of 32 tanh units. The other tasks make use of a two layer neural network policy of 32 tanh units each for MountainCar and HalfCheetah, and of 64 and 32 tanh units for SwimmerGather. The outputs are modeled by a fully factorized Gaussian distribution $\mathcal{N}(\mu, \sigma^2 I)$, in which $\mu$ is modeled as the network output, while $\sigma$ is a parameter. CartPoleSwingup makes use of a neural network baseline with one layer of 32 ReLU units, while all other tasks make use of a linear baseline function. For all tasks, we used TRPO step size 0.01 and discount factor $\gamma = 0.99$. We choose SimHash parameter $k = 32$ and bonus coefficient $\beta = 0.01$, found through a coarse grid search.

For Atari experiments, a batch size of 100000 is used, while the KL divergence step size is set to 0.01. The policy and baseline both have the following architecture: 2 convolutional layers with respectively 16 and 32 filters, sizes $8 \times 8$ and $4 \times 4$, strides 4 and 2, using no padding, feeding into a single hidden layer of 256 units. The nonlinearities are rectified linear units (ReLUs). The input frames are downsampled to $52 \times 52$. The input to policy and baseline consists of the 4 previous frames, corresponding to the frame skip of 4. The discount factor was set to $\gamma = 0.995$. All inputs are rescaled to $[-1, 1]$ element-wise. All experiments used 5 different training seeds, except the experiments with the learned hash code, which uses 3 different training seeds. Batch normalization (Ioffe & Szegedy, 2015) is used at each policy and baseline layer. TRPO-pixel-SimHash uses binary codes of size $k = 256$; BASS (TRPO-BASS-SimHash) extracts features using cell size $C = 20$ and $B = 20$ bins. The autoencoder for the learned embedding (TRPO-AE-SimHash) uses a binary hidden layer of 512 bit, which are projected to 64 bit.

RAM states in Atari 2600 games are integer-valued vectors over length 128 in the range $[0, 255]$. Experiments on Montezuma's Revenge with RAM observations use a policy consisting of 2 hidden layers, each of size 32. RAM states are rescaled to a range $[-1, 1]$. Unlike images, only the current RAM is shown to the agent. Experiment results are averaged over 10 random seeds.

In addition, we apply counting Bloom filters (Fan et al., 2000) to maintain a small hash table. Details can be found in Appendix A.5.

The autoencoder used for the learned hash code has a 512 bit binary code layer, using sigmoid units, to which uniform noise $U(-a, a)$ with $a = 0.3$ is added. The loss function Eq. (3), using $\lambda = 10$, is updated every $j_{\text{update}} = 3$ iterations. The architecture looks as follows: an input layer of size $52 \times 52$, representing the image luminance is followed by 3 consecutive $6 \times 6$ convolutional layers with stride 2 and 96 filters feed into a fully connected layer of size 1024, which connects to the binary code layer. This binary code layer feeds into a fully-connected layer of 1024 units, connecting to a fully-connected layer of 2400 units. This layer feeds into 3 consecutive $6 \times 6$ transposed convolutional layers of which the final one connects to a pixel-wise softmax layer with 64 bins, representing the pixel intensities. Moreover, label smoothing is applied to the different softmax bins, in which the log-probability of each of the bins is increased by 0.003, before normalizing. The softmax weights are shared among each pixel. All output nonlinearities are ReLUs; Adam (Kingma & Ba, 2015) is used as an optimization scheme; batch normalization (Ioffe & Szegedy, 2015) is applied to each layer. The architecture was shown in Figure 1 of Section 2.3.

## A.2  DESCRIPTION OF THE ADAPTED RLLAB TASKS

This section describes the continuous control environments used in the experiments. The tasks are implemented as described in Duan et al. (2016), following the sparse reward adaptation of Houthooft et al. (2016). The tasks have the following state and action dimensions: CartPoleSwingup, $\mathcal{S} \subseteq \mathbb{R}^4$, $\mathcal{A} \subseteq \mathbb{R}^1$; MountainCar $\mathcal{S} \subseteq \mathbb{R}^3$, $\mathcal{A} \subseteq \mathbb{R}^1$; HalfCheetah, $\mathcal{S} \subseteq \mathbb{R}^{20}$, $\mathcal{A} \subseteq \mathbb{R}^6$; SwimmerGather, $\mathcal{S} \subseteq \mathbb{R}^{33}$, $\mathcal{A} \subseteq \mathbb{R}^2$. For the sparse reward experiments, the tasks have been modified as follows. In CartPoleSwingup, the agent receives a reward of +1 when $\cos(\beta) > 0.8$, with $\beta$ the pole angle. In MountainCar, the agent receives a reward of +1 when the goal state is reached, namely escaping the valley from the right side. Therefore, the agent has to figure out how to swing up the pole in the absence of any initial external rewards. In HalfCheetah, the agent receives a reward of +1 when

$x_{\text{body}} > 5$. As such, it has to figure out how to move forward without any initial external reward. The time horizon is set to $T = 500$ for all tasks.

## A.3 Examples of Atari 2600 RAM Entries

Table 5 lists the semantic interpretation of certain RAM entries in Montezuma's Revenge. SmartHash, as described in Section 3.4, makes use of RAM indices 3, 42, 43, 27, and 67. "Beam walls" are deadly barriers that occur periodically in some rooms.

Table 5: Interpretation of particular RAM entries in Montezuma's Revenge

| RAM index | Group | Meaning |
|---|---|---|
| 3 | room | room number |
| 42 | agent | $x$ coordinate |
| 43 | agent | $y$ coordinate |
| 52 | agent | orientation (left/right) |
| 27 | beam walls | on/off |
| 83 | beam walls | beam wall countdown (on: 0, off: $36 \rightarrow 0$) |
| 0 | counter | counts from 0 to 255 and repeats |
| 55 | counter | death scene countdown |
| 67 | objects | existence of objects (doors, skull and key) in the 1st room |
| 47 | skull | $x$ coordinate (both 1st and 2nd rooms) |

## A.4 Analysis of Learned Binary Representation

Figure 6 shows the downsampled codes learned by the autoencoder for several Atari 2600 games (Frostbite, Freeway, and Montezuma's Revenge). Each row depicts 50 consecutive frames (from 0 to 49, going from left to right, top to bottom). The pictures in the right column depict the binary codes that correspond with each of these frames (one frame per row). Figure 7 shows the reconstructions of several subsequent images according to the autoencoder.

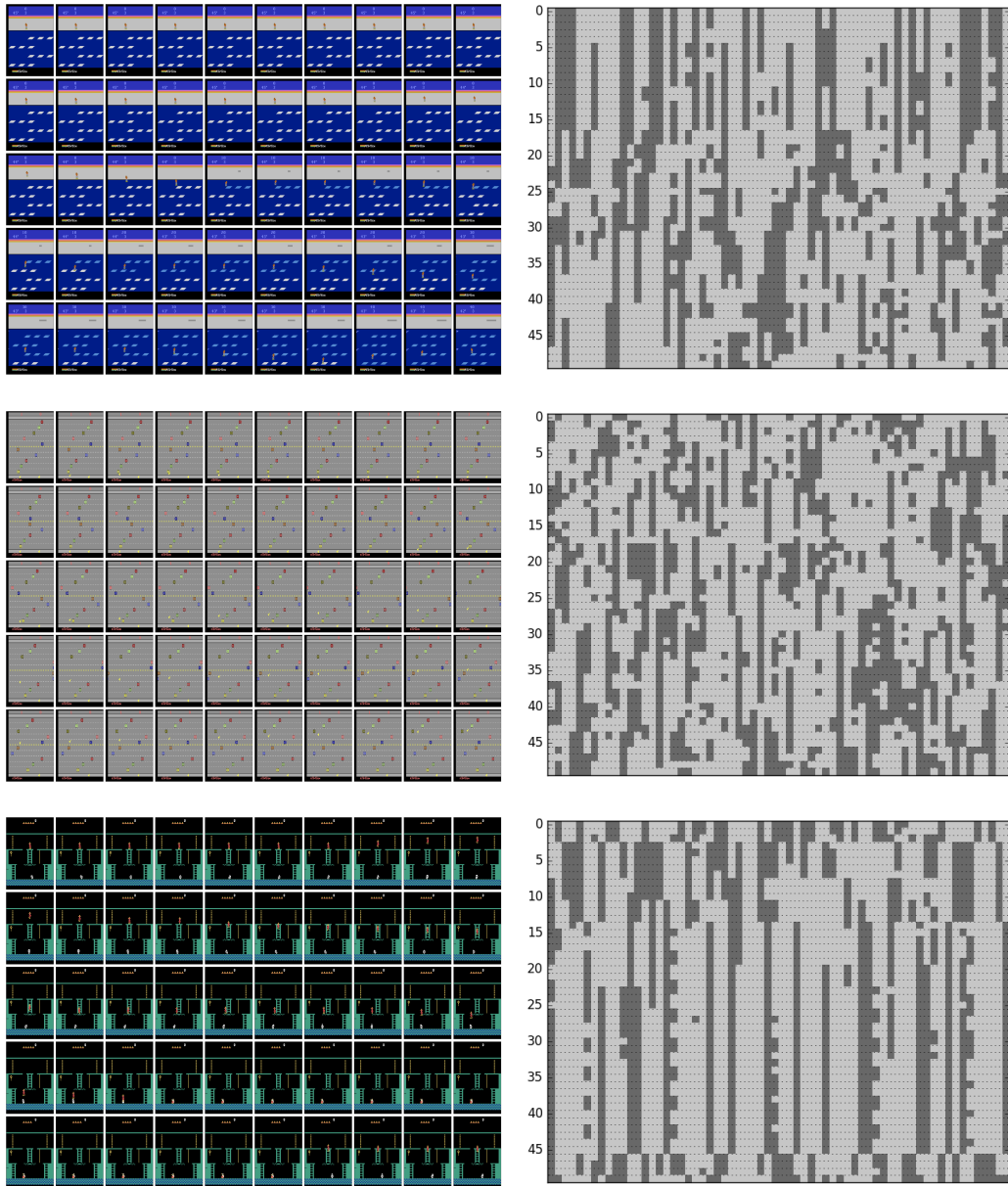

Figure 6: Frostbite, Freeway, and Montezuma's Revenge: subsequent frames (left) and corresponding code (right); the frames are ordered from left (starting with frame number 0) to right, top to bottom; the vertical axis in the right images correspond to the frame number.

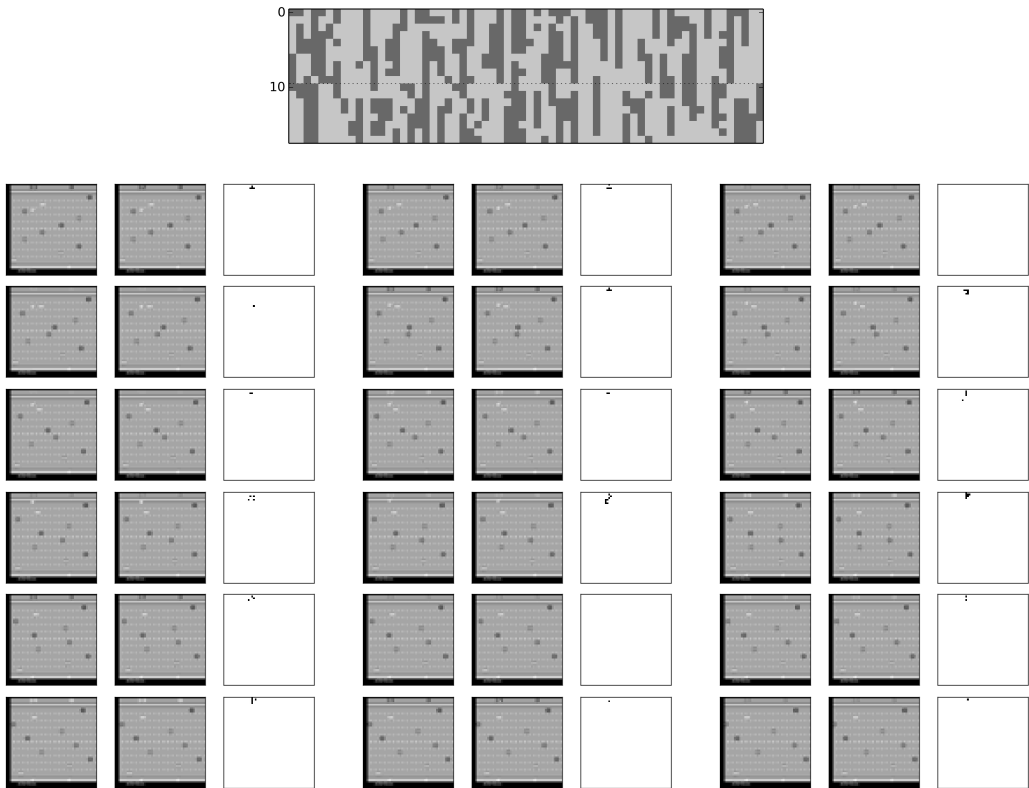

Figure 7: Freeway: subsequent frames and corresponding code (top); the frames are ordered from left (starting with frame number 0) to right, top to bottom; the vertical axis in the right images correspond to the frame number. Within each image, the left picture is the input frame, the middle picture the reconstruction, and the right picture, the reconstruction error.

## A.5 COUNTING BLOOM FILTER/COUNT-MIN SKETCH

We experimented with directly building a hashing dictionary with keys $\phi(s)$ and values the state counts, but observed an unnecessary increase in computation time. Our implementation converts the integer hash codes into binary numbers and then into the "bytes" type in Python. The hash table is a dictionary using those bytes as keys.

However, an alternative technique called Count-Min Sketch (Cormode & Muthukrishnan, 2005), with a data structure identical to counting Bloom filters (Fan et al., 2000), can count with a fixed integer array and thus reduce computation time. Specifically, let $p^1, \ldots, p^l$ be distinct large prime numbers and define $\phi^j(s) = \phi(s) \mod p^j$. The count of state $s$ is returned as $\min_{1 \le j \le l} n^j\left(\phi^j(s)\right)$. To increase the count of $s$, we increment $n^j\left(\phi^j(s)\right)$ by 1 for all $j$. Intuitively, the method replaces $\phi$ by weaker hash functions, while it reduces the probability of over-counting by reporting counts agreed by all such weaker hash functions. The final hash code is represented as $\left(\phi^1(s), \ldots, \phi^l(s)\right)$.

Throughout all experiments above, the prime numbers for the counting Bloom filter are 999931, 999953, 999959, 999961, 999979, and 999983, which we abbreviate as "6 M". In addition, we experimented with 6 other prime numbers, each approximately 15 M, which we abbreviate as "90 M". As we can see in Figure 8, counting states with a dictionary or with Bloom filters lead to similar performance, but the computation time of latter is lower. Moreover, there is little difference between direct counting and using a very larger table for Bloom filters, as the average bonus rewards are almost the same, indicating the same degree of exploration-exploitation trade-off. On the other hand, Bloom filters require a fixed table size, which may not be known beforehand.

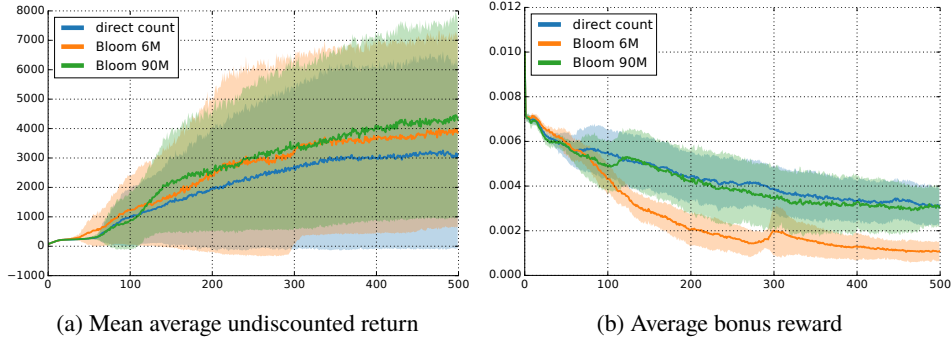

| (a) Mean average undiscounted return | (b) Average bonus reward |

Figure 8: Statistics of TRPO-pixel-SimHash ($k = 256$) on Frostbite. Solid lines are the mean, while the shaded areas represent the one standard deviation. Results are derived from 10 random seeds. Direct counting with a dictionary uses 2.7 times more computations than counting Bloom filters (6 M or 90 M).

**Theory of Bloom Filters**   Bloom filters (Bloom, 1970) are popular for determining whether a data sample $s'$ belongs to a dataset $\mathcal{D}$. Suppose we have $l$ functions $\phi^j$ that independently assign each data sample to an integer between 1 and $p$ uniformly at random. Initially $1, 2, \ldots, p$ are marked as 0. Then every $s \in \mathcal{D}$ is "inserted" through marking $\phi^j(s)$ as 1 for all $j$. A new sample $s'$ is reported as a member of $\mathcal{D}$ only if $\phi^j(s)$ are marked as 1 for all $j$. A bloom filter has zero false negative rate (any $s \in \mathcal{D}$ is reported a member), while the false positive rate (probability of reporting a nonmember as a member) decays exponentially in $l$.

Though Bloom filters support data insertion, it does not allow data deletion. Counting Bloom filters (Fan et al., 2000) maintain a counter $n(\cdot)$ for each number between 1 and $p$. Inserting/deleting $s$ corresponds to incrementing/decrementing $n\big(\phi^j(s)\big)$ by 1 for all $j$. Similarly, $s$ is considered a member if $\forall j : n\big(\phi^j(s)\big) = 0$.

Count-Min sketch is designed to support memory-efficient counting without introducing too many over-counts. It maintains a separate count $n^j$ for each hash function $\phi^j$ defined as $\phi^j(s) = \phi(s) \mod p^j$, where $p^j$ is a large prime number. For simplicity, we may assume that $p^j \approx p \; \forall j$ and $\phi^j$ assigns $s$ to any of $1, \ldots, p$ with uniform probability.

We now derive the probability of over-counting. Let $s$ be a fixed data sample (not necessarily inserted yet) and suppose a dataset $\mathcal{D}$ of $N$ samples are inserted. We assume that $p^l \gg N$. Let $n := \min_{1 \leq j \leq l} n^j\big(\phi^j(s)\big)$ be the count returned by the Bloom filter. We are interested in computing $\mathrm{Prob}(n > 0 | s \notin \mathcal{D})$. Due to assumptions about $\phi^j$, we know $n^j(\phi(s)) \sim \mathrm{Binomial}\left(N, \frac{1}{p}\right)$. Therefore,

$$
\begin{aligned}
\mathrm{Prob}(n > 0 | s \notin \mathcal{D}) &= \frac{\mathrm{Prob}(n > 0, s \notin \mathcal{D})}{\mathrm{Prob}(s \notin \mathcal{D})} \\
&= \frac{\mathrm{Prob}(n > 0) - \mathrm{Prob}(s \in \mathcal{D})}{\mathrm{Prob}(s \notin \mathcal{D})} \\
&\approx \frac{\mathrm{Prob}(n > 0)}{\mathrm{Prob}(s \notin \mathcal{D})} \\
&= \frac{\prod_{j=1}^{l} \mathrm{Prob}(n^j(\phi^j(s)) > 0)}{(1 - 1/p^l)^N} \\
&= \frac{(1 - (1 - 1/p)^N)^l}{(1 - 1/p^l)^N} \\
&\approx \frac{(1 - e^{-N/p})^l}{e^{-N/p^l}} \\
&\approx (1 - e^{-N/p})^l.
\end{aligned}
\tag{5}
$$

In particular, the probability of over-counting decays exponentially in $l$. We refer the readers to (Cormode & Muthukrishnan, 2005) for other properties of the Count-Min sketch.

A.6 ROBUSTNESS ANALYSIS

Apart from the experimental results shown in Table 1 and Table 3, additional experiments have been performed to study several properties of our algorithm.

**Hyperparameter sensitivity** To study the performance sensitivity to hyperparameter changes, we focus on evaluating TRPO-RAM-SimHash on the Atari 2600 game Frostbite, where the method has a clear advantage over the baseline. Because the final scores can vary between different random seeds, we evaluated each set of hyperparameters with 30 seeds. To reduce computation time and cost, RAM states are used instead of image observations.

Table 6: TRPO-RAM-SimHash performance robustness to hyperparameter changes on Frostbite

| $k$ | $\beta$ | | | | | | | |
|---|---|---|---|---|---|---|---|---|
| | 0 | 0.01 | 0.05 | 0.1 | 0.2 | 0.4 | 0.8 | 1.6 |
| – | 397 | – | – | – | – | – | – | – |
| 64 | – | 879 | 2464 | 2243 | 2489 | 1587 | 1107 | 441 |
| 128 | – | 1475 | 4248 | 2801 | 3239 | 3621 | 1543 | 395 |
| 256 | – | 2583 | 4497 | 4437 | 7849 | 3516 | 2260 | 374 |

The results are summarized in Table 6. Herein, $k$ refers to the length of the binary code for hashing while $\beta$ is the multiplicative coefficient for the reward bonus, as defined in Section 2.2. This table demonstrates that most hyperparameter settings outperform the baseline ($\beta = 0$) significantly. Moreover, the final scores show a clear pattern in response to changing hyperparameters. Small $\beta$-values lead to insufficient exploration, while large $\beta$-values cause the bonus rewards to overwhelm the true rewards. With a fixed $k$, the scores are roughly concave in $\beta$, peaking at around 0.2. Higher granularity $k$ leads to better performance. Therefore, it can be concluded that the proposed exploration method is robust to hyperparameter changes in comparison to the baseline, and that the best parameter settings can obtained from a relatively coarse-grained grid search.

**State and state-action counting** Continuing the results in Table 6, the performance of state-action counting is studied using the same experimental setup, summarized in Table 7. In particular, a bonus reward $r^+ = \frac{\beta}{\sqrt{n(s,a)}}$ instead of $r^+ = \frac{\beta}{\sqrt{n(s)}}$ is assigned. These results show that the relative performance of state counting compared to state-action counting depends highly on the selected hyperparameter settings. However, we notice that the best performance is achieved using state counting with $k = 256$ and $\beta = 0.2$.

Table 7: Performance comparison between state counting (left of the slash) and state-action counting (right of the slash) using TRPO-RAM-SimHash on Frostbite

| $k$ | $\beta$ | | | | | | |
|---|---|---|---|---|---|---|---|
| | 0.01 | 0.05 | 0.1 | 0.2 | 0.4 | 0.8 | 1.6 |
| 64 | 879 / 976 | 2464 / 1491 | 2243 / 3954 | 2489 / 5523 | 1587 / 5985 | 1107 / 2052 | 441 / 742 |
| 128 | 1475 / 808 | 4248 / 4302 | 2801 / 4802 | 3239 / 7291 | 3621 / 4243 | 1543 / 1941 | 395 / 362 |
| 256 | 2583 / 1584 | 4497 / 5402 | 4437 / 5431 | 7849 / 4872 | 3516 / 3175 | 2260 / 1238 | 374 / 96 |

