# Peer review of "#Exploration: A Study of Count-Based Exploration for Deep Reinforcement Learning"

_ICLR 2017 — rejected_

[Public Comment · Marc G Bellemare · 18 Nov 2016]
**Is this a hash-based density model?**

It's very exciting to see more work towards cracking Montezuma's Revenge. I think the point that bonus-based algorithms might "focus on trivial details" is quite relevant -- this will clearly be a fundamental issue of exploration going forward. Showing that count generalization can be done with LSH is also quite valuable. A few comments:

1) Rather than try to separate the hash-table approach from the density model approach, I want to view the two as doing similar things. The LSH + counting scheme implies a density model of the form n(c_t) / total_count. So one might view your approach as using a hash-based density model, which I think is really cool!

2) On the hashing front: I would have expected a set of hash tables, as is typically done in the sketches literature, to work better. Did you consider approaches such as the count-min sketch?

3) It might be worth pointing out that the scores in Table 1 aren't completely comparable, as some very relevant parameters differ: the evaluation epsilons, the type of stochastic perturbation involved, etc. This shouldn't detract from the results, since your approach is clearly outperforming the no-smart-exploration baseline.

4) The Montezuma's Revenge for "pseudo-counts" are at 100M frames (25M agent steps), rather than 200M, so it isn't quite comparable -- again, might be worth pointing out if anyone grabs the score from your table. For reference, here are the other scores at 100M frames: Freeway 29.22, Frostbite 1450, Montezuma's Revenge 3439, Venture 369.

5) Did you look at the learned policy for Solaris? I'd be curious to see the agent's behaviour.

[Official Review · AnonReviewer2 · rating 7 · confidence 4 · 17 Dec 2016]
**Solid paper**

This paper proposed to use a simple count-based exploration technique in high-dimensional RL application (e.g., Atari Games). The counting is based on state hash, which implicitly groups (quantizes) similar state together. The hash is computed either via hand-designed features or learned features (unsupervisedly with auto-encoder). The new state to be explored receives a bonus similar to UCB (to encourage further exploration).

Overall the paper is solid with quite extensive experiments. I wonder how it generalizes to more Atari games. Montezuma’s Revenge may be particularly suitable for approaches that implicitly/explicitly cluster states together (like the proposed one), as it has multiple distinct scenarios, each with small variations in terms of visual appearance, showing clustering structures. On the other hand, such approaches might not work as well if the state space is fully continuous (e.g. in RLLab experiments). 

The authors did not answer my question about why the hash code needs to be updated during training. I think it is mainly because the code still needs to be adaptive for a particular game (to achieve lower reconstruction error) in the first few iterations . After that stabilization is the most important. Sec. 2.3 (Learned embedding) is quite confusing (but very important). I hope that the authors could make it more clear (e.g., by writing an algorithm block) in the next version.

[Official Review · AnonReviewer1 · rating 6 · confidence 4 · 19 Dec 2016 (modified: 23 Jan 2017)]
**Final review: significant results in an important problem, but many moving parts**

The paper proposes a new exploration scheme for reinforcement learning using locality-sensitive hashing states to build a table of visit counts which are then used to encourage exploration in the style of MBIE-EB of Strehl and Littman.

Several points are appealing about this approach: first, it is quite simple compared to the current alternatives (e.g. VIME, density estimation and pseudo-counts). Second, the paper presents results across several domains, including classic benchmarks, continuous control domains, and Atari 2600 games. In addition, there are results for comparison from several other algorithms (DQN variants), many of which are quite recent. The results indicate that the approach clearly improves over the baseline. The results against other exploration algorithms are not as clear (more dependent on the individual domain/game), but I think this is fine as the appeal of the technique is its simplicity. Third, the paper presents results on the sensitivity to the granularity of the abstraction.

I have only one main complaint, which is it seems there was some engineering involved to get this to work, and I do not have much confidence in the robustness of the conclusions. I am left uncertain as to how the story changes given slight perturbations over hyper-parameter values or enabling/disabling of certain choices. For example, how critical was using PixelCNN (or tying the weights?) or noisifying the output in the autoencoder, or what happens if you remove the custom additions to BASS? The granularity results show that the choice of resolution is sensitive, and even across games the story is not consistent.

The authors decide to use state-based counts instead of state-action based counts, deviating from the theory, which is odd because the reason to used LSH in the first place is to get closer to what MBIE-EB would advise via tabular counts. There are several explanations as to why state-based versus state-action based counts perform similarly in Atari; the authors do not offer any. Why?

It seems like the technique could be easily used in DQN as well, and many of the variants the authors compare to are DQN-based, so omitting DQN here again seems strange. The authors justify their choice of TRPO by saying it ensures safe policy improvement, though it is not clear that this is still true when adding these exploration bonuses.

The case study on Montezuma's revenge, while interesting, involves using domain knowledge and so does not really fit well with the rest of the paper.

So, in the end, simple and elegant idea to help with exploration tested in many domains, though I am not certain which of the many pieces are critical for the story to hold versus just slightly helpful, which could hurt the long-term impact of the paper.

--- After response:

Thank you for the thorough response, and again my apologies for the late reply.

I appreciate the follow-up version on the robustness of SimHash and state counting vs. state-action counting.

The paper addresses an important problem (exploration), suggesting a "simple" (compared to density estimation) counting method via hashing. It is a nice alternative approach to the one offered by Bellemare et al. If discussion among reviewers were possible, I would now try to assemble an argument to accept the paper. Specifically, I am not as concerned about beating the state of the art in Montezuma's as Reviewer3 as the merit of the current paper is one the simplicity of the hashing and on the wide comparison of domains vs. the baseline TRPO. This paper shows that we should not give up on simple hashing. There still seems to be a bunch of fiddly bits to get this to work, and I am still not confident that these results are easily reproducible. Nonetheless, it is an interesting new contrasting approach to exploration which deserves attention.

Not important for the decision: The argument in the rebuttal concerning DQN & A3C is a bit of a straw man. I did not mention anything at all about A3C, I strictly referred to DQN, which is less sensitive to parameter-tuning than A3C. Also, Bellemare 2016 main result on Montezuma used DQN. Hence the omission of these techniques applied to DQN still seems a bit strange (for the Atari experiments). The figure S9 from Mnih et al. points to instances of asynchronous one-step Sarsa with varied thread counts.. of course this will be sensitive to parameters: it is both asynchronous online algorithms *and* the parameter varied is the thread count! This is hardly indicative of DQN's sensitivity to parameters, since DQN is (a) single-threaded (b) uses experience replay, leading to slower policy changes. Another source of stability, DQN uses a target network that changes infrequently. Perhaps the authors made a mistake in the reference graph in the figure? (I see no Figure 9 in

[Official Review · AnonReviewer3 · rating 4 · confidence 3 · 22 Dec 2016]

This paper introduces a new way of extending the count based exploration approach to domains where counts are not readily available. The way in which the authors do it is through hash functions. Experiments are conducted on several domains including control and Atari. 

It is nice that the authors confirmed the results of Bellemare in that given the right "density" estimator, count based exploration can be effective. It is also great the observe that given the right features, we can crack games like Montezuma's revenge to some extend.

I, however, have several complaints:

First, by using hashing, the authors did not seem to be able to achieve significant improvements over past approaches. Without "feature engineering", the authors achieved only a fraction of the performance achieved in Bellemare et al. on Montezuma's Revenge. The proposed approaches In the control domains, the authors also does not outperform VIME. So experimentally, it is very hard to justify the approach. 

Second, hashing, although could be effective in the domains that the authors tested on, it may not be the best way of estimating densities going forward. As the environments get more complicated, some learning methods, are required for the understanding of the environments instead of blind hashing. The authors claim that the advantage of the proposed method over Bellemare et al. is that one does not have to design density estimators. But I would argue that density estimators have become readily available (PixelCNN, VAEs, Real NVP, GANs) that they can be as easily applied as can hashing. Training the density estimators is not difficult problem as more.

[Author Response · Haoran Tang · 09 Jan 2017]
**Paper revision**

We have updated the paper with the following changes

(1) The distinction between static hashing (e.g. LSH) and learned hashing is made clear. They are now discussed separately in sections 2.2 and 2.3. BASS is moved to the experimental part (section 3.2), since it is specific to Atari.

(2) The text in section 2.3 is improved to clarify details of training the autoencoder. A separate algorithm block (Algorithm 2) clarifies that the autoencoder is retrained periodically.

(3) Robustness of SimHash and and the comparison between state counting and state action counting are included in Appendix A.6.

[Final Decision · Program Chairs · 06 Feb 2017]
**ICLR committee final decision**

The paper proposes a simple approach to exploration that uses a hash of the current state within a exploration bonus approach (there are some modifications to learned hash codes, but this is the basic approach). The method achieves reasonable performance on Atari game tasks (sometimes outperformed by other approaches, but overall performing well), and it's simplicity is its main appeal (although the autoencoder-based learned hash seems substantially less simple, so actually loses some advantage there). 
 
 The paper is likely borderline, as the results are not fantastic: the approach is typically outperformed or similarly-performed by one of the comparison approaches (though it should be noted that no comparison approach performs well over all tasks, so this is not necessarily that bad). But overall, especially because so many of these methods have tunable hyperparameters it was difficult to get a clear understanding of just how these experimental results fit.
 
 Pros:
 + Simple method for exploration that seems to work reasonably well in practice
 + Would have the potential to be widely used because of its simplicity
 
 Cons:
 - Improvements over previous approaches are not always there, and it's not clear whether the algorithm has any "killer app" domain where it is just clearly the best approach
  
 Overall, this work in its current form is too borderline. The PCs encourage the authors to strengthen the empirical validation and resubmit.